# The effects of five weeks of climbing training, on and off the wall, on climbing specific strength, performance, and training experience in female climbers—A randomized controlled trial

Kaja Langer[1]*, Vidar Andersen[2], Nicolay Stien[2]

1 Institute of Sports Science, Technical University of Darmstadt, Darmstadt, Germany, 2 Faculty of Education, Arts and Sports, Western Norway University of Applied Sciences, Sogndal, Bergen, Norway

* kaja.langer@tu-darmstadt.de

**Data Availability Statement:** All relevant data are within the paper and its Supporting Information files.

## Abstract

Recent research has elucidated the effects of strength training on climbing performance. Although local muscular endurance training of the upper-limbs and finger flexors is frequently suggested, there is currently insufficient evidence to support its impact on climbing performance and climbing-specific strength. Furthermore, there is no evidence on climbers' experiences related to training and the likelihood of consistent engagement. In addition, the effects of more climbing-specific strength training on walls with built in lights and adjustable angles have yet to be examined. The low percentage of studies involving female subjects, additionally, demonstrates a significant gap in understanding the specific effects of strength training on women in the context of climbing. The aim of this study was thus to assess the effects of five-week on-, and off-the-wall climbing training on climbing performance, climbing-specific strength, and training experience. Thirty-one female lower-grade to advanced climbers were randomly assigned to either a control group, an off-the-wall training or an on-the-wall training group. Apart from the training regimen, all groups followed their usual climbing and bouldering routine. Subjects trained at least twice a week. Bouldering performance, and maximum strength and muscular endurance of the finger flexors and upper-limbs were assessed before and after the intervention. Furthermore, rate of perceived exertion and discomfort, exercise enjoyment, and exercise pleasure were assessed during the first and last training session, as well as after two and a half weeks of training. Intrinsic training motivation was assessed after the last training session. The results showed trends towards positive effects of off-the-wall training on climbing-specific strength, and on-the-wall training on climbing technique. Furthermore, our finding revealed high exercise enjoyment and intrinsic training motivation for both on- and off-the-wall training. Hence, lower-grade to advanced female climbers should rely on personal training preferences.

**Funding:** The author(s) received no specific funding for this work.

**Competing interests:** There were no professional relationships with companies or manufacturers that will benefit from the results of this study. There was no specific financial support for the preparation of this study. The authors have no potential conflicts of interest that are directly related to the study and the journal.

## Introduction

In recent years, climbing has evolved from a trend sport, practiced by only few people, to a common recreational sport. This development was particularly promoted by climbing becoming an Olympic Sport in 2020(21) [1] and is accompanied by a professionalization and expansion of the sport. This has led to an increasing number of athletes and coaches looking for ways to improve climbing performance both in professional and amateur contexts.

Multiple studies have shown that strength is a key factor of climbing performance. Especially maximal finger strength [2–5], isometric muscular endurance of the finger flexors [5, 6] upper-limb power [5], and shoulder girdle muscular endurance [3] were found to highly correlate with climbing performance.

Consequently, it is advised to engage in prolonged participation in conjunction with climbing-specific strength training to ensure progress in climbing and climbing-specific factors, and to lower the susceptibility to injuries [6–11].

The effectiveness of certain training exercises and methods in improving climbing performance has been partly demonstrated [8, 7]. However, the state of research is limited. The effects of muscular endurance training, for example, have been investigated by only very few studies [12, 13]. This is despite the fact that especially muscular endurance training with low intensities is recommended for lower-level climbers and climbers without experience in climbing-specific training [11, 14]. In two studies, Hermans et al. [13, 15] reported that muscular endurance training including non-climbing-specific exercises such as pull-down, seated bench press, seated rowing, seated shoulder press, biceps curl, forearm press, and forearm curl leads to an improvement in climbing-specific tests but not in climbing performance. A review by Langer et al. [8], suggests that unlike unspecific training, exercises more closely related to the climbing movement might lead to larger positive effects in climbing performance. This hypothesis is supported by the findings of multiple studies, suggesting to combine strength training with technical aspects in order to increase sports performance [6, 16–20]. Philippe et al. [16] demonstrated improved lead climbing performance after eight weeks of specific muscular endurance training through bouldering and lead climbing in advanced climbers. In addition, a study by Medernach et al. [21] suggested muscular endurance training through interval bouldering to be a highly effective method to increase upper-limb muscular endurance in elite boulder athletes. Furthermore, Stien et al. [22] observed and increase in isometric pull-up strength, and isolated finger strength after five weeks of prioritized bouldering. Fryer et al. [23] on the other hand suggested intermittent fatigue protocols on the fingerboard, especially for lower-level climbers. The effectiveness of semi-specific off-the-wall training has not been compared with on-the-wall training in any study to date. Additionally, no research has yet examined the impact of muscular endurance training on athletes at lower to advanced levels.

Furthermore, there remains a notable gap in the understanding of climbers' perceptions and experiences related to climbing-specific training and the likelihood of consistent engagement, especially in amateur sports. Emotional and motivational responses, however, are reported to be increasingly important in the context of long-term engagement in sports and sport-specific training [24–26]. Lack of enjoyment, and lack of time for example have been found to hinder participation in regular training activities and thus impair the training progress [27, 28]. Other factors that have been identified as barriers to training, especially for women, include gender stigma, negative comments, discouragement, boredom, and lack of knowledge about resistance training [29]. According to Vasudevan and Ford [29], and Chevan [30] women are thus less likely to participate in resistance training than men. Enjoyment, comfort, intrinsic motivation, and perceived usefulness, on the other hand, have been shown to have a strong positive influence on the willingness to continue an activity over time for both

men and women [27, 31, 32]. It is therefore of great interest to investigate how female athletes, in particular, experience different types of climbing training.

Furthermore, it is important to note that research on the effects of resistance training on climbing performance has predominantly focused on male subjects [7, 33, 34]. This focus persists despite the fact that male and female athletes have demonstrated comparable levels of performance. In addition, it has been reported that female non-top-level climbers in particular express a performance and strength deficit compared to top-level climbers [35], and thus may particularly profit from additional training [34].

The aim of our study was to assess the effects of muscular endurance training on- and off-the-wall in lower-grade to advanced female climbers. The following research hypothesis (RH) were tested:

- RH1: Bouldering performance improves in both intervention groups but not in CG.

- RH2: ST improves more in climbing specific strength compared to WT and CG.

- RH3: WT improves more in climbing technique compared to ST and CG.

- RH4: Rate of perceived exertion and discomfort are consistent over the course of the study and similar in both intervention groups.

- RH5: Exercise enjoyment, pleasure, and intrinsic training motivation are higher in WT compared to ST, and exercise enjoyment and pleasure are consistent over the course of the study.

## Materials and methods

### Study design

The study involved lower-grade to advanced female climbers who underwent a five-week training intervention comprising either off-the-wall muscular endurance training (ST), or technical oriented on-the-wall boulder training (WT). Training effects were compared to a control group (CG), who followed their usual climbing routine for the course of the study. The study design corresponds to a randomized controlled format [36]. Thirty-one participants were randomly assigned to ST, WT or CG after pre-testing by pulling a piece of paper with ST, WT, or CG out of a hat. In order to allow for sufficient study power when utilizing Bayesian statistics (80%, BF $\geq$ 10), Brysbaert [37] recommends at least 52 participants for 2 x 2 hypothesis testing. Due to limited participant availability, we were only able to recruit thirty-one participants. While all participants were instructed to stick to their usual climbing and bouldering routines throughout the study period, individuals in the ST and WT group incorporated a 30-minute training program into their climbing practice. The training was not added to their existing routines but replaced a portion of their regular training volume. Subjects in the intervention groups had to conduct the intervention sessions two times per week, adding up to ten sessions, but were allowed to increase the number of sessions if motivated. An intervention duration of five weeks was chosen in order to allow for sufficient adaptions to muscular endurance training to occur, as recommended by Schmidtbleicher [38] [Classification of strength and strength training methods differ between German and English publications which is why a German reference was chosen. Similar recommendations can be found here: [39, 40]]. Two sessions per week were implemented in order to incorporate targeted training methods into the participants' existing routines rather than adding to their training volume.

Pre- and post-test included four strength tests: a) isometric pull up, b) bent arm hang, c) maximum finger strength test, d) and dead hang, in addition to performance testing on five boulder problems. Emotional measurements (exercise enjoyment, exercise comfort, and

exercise pleasure) were conducted after the first, a middle and the last training session. Motivational measurements were conducted after the last training session. Prior to the pre-test all subjects participated in a familiarization session during which their anthropometrics were recorded and they were familiarized with all strength tests.

## Subjects

Subjects had to be female and at least 18 years old. Furthermore, they were required to have engaged in a regular, bouldering activity over the last three months (at least twice a week). On-sight level of the subjects was not allowed to be higher than 6b+ on the Fontainebleau scale (International Rock Climbing Research Association (IRCRA) scale 18), in order to include only lower grade to advanced climbers [41]. Recruitment occurred in the two weeks leading up to the study (03/10/2022–16/10/2022), utilizing flyers distributed in campus climbing facilities and personal outreach by the researchers during seminars and at the climbing facilities. Thirty-one lower level to advanced female climbers (active in bouldering and climbing) volunteered for the study. They were all healthy and free of injury for the past three months. During the five-week intervention, five subjects experienced an injury unrelated to the study which left a total of twenty-six subjects finishing all training and testing procedures. A flow diagram on the progress through the phases is presented in Fig 1. While most of the participants were familiar with the fingerboard, none of them had ever used it for regular training purposes. Furthermore, no participant had ever been involved in any climbing-specific training. Sample characteristics are presented in Table 1. Climbing grades were converted according the scale introduced by the IRCRA [41].

## Ethics statement

The study was approved by the Regional Committees for Medical Health and Research Ethics in Norway (992074) and conformed to standards of treatment of human participants in

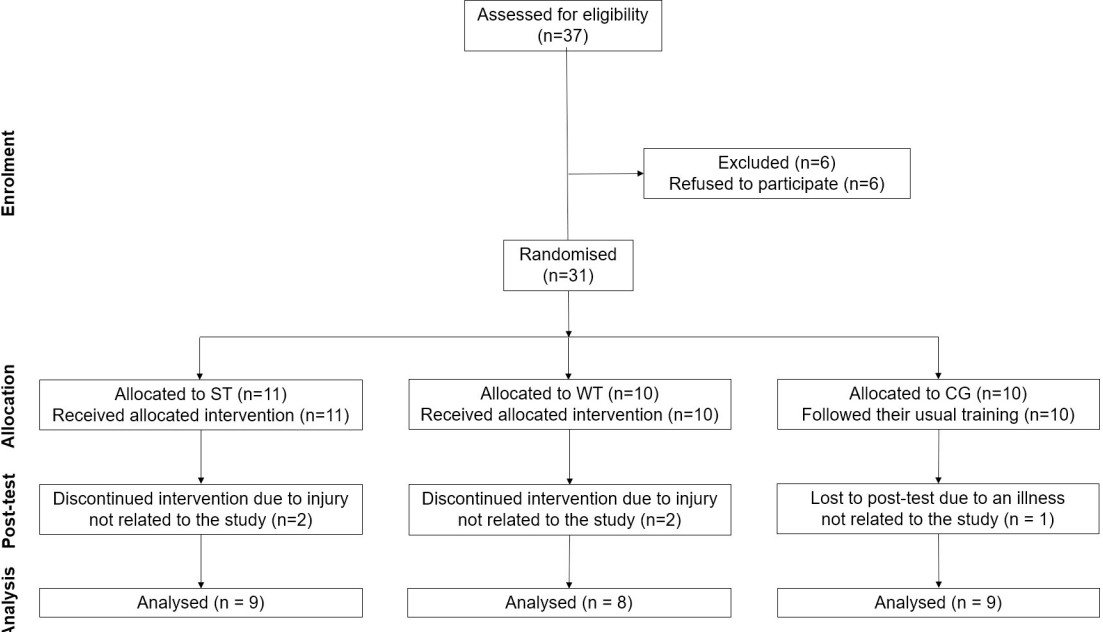

**Fig 1. Flow diagram of the progress through the phases (CONSORT diagram) as proposed by Moher et al. [42].**

**Table 1. Sample characteristics by group.**

| Factor | Group | N | Mean | Std. Deviation | Minimum | Maximum |
|---|---|---|---|---|---|---|
| Age [yrs] | CG | 9 | 23.67 | 3.67 | 20 | 32 |
| | ST | 9 | 24.22 | 3.03 | 19 | 28 |
| | WT | 8 | 24.13 | 3.09 | 21 | 31 |
| | Total | 26 | 24.00 | 3.16 | 19 | 32 |
| Height [cm] | CG | 9 | 167.67 | 2.12 | 165.00 | 171.00 |
| | ST | 9 | 169.89 | 2.98 | 165.00 | 174.00 |
| | WT | 8 | 168.88 | 8.59 | 163.00 | 188.00 |
| | Total | 26 | 168.81 | 5.08 | 163.00 | 188.00 |
| Weight [kg] | CG | 9 | 67.03 | 5.16 | 59.80 | 76.00 |
| | ST | 9 | 62.72 | 3.96 | 55.50 | 68.30 |
| | WT | 8 | 64.46 | 13.46 | 47.70 | 89.10 |
| | Total | 26 | 64.75 | 8.23 | 47.70 | 89.10 |
| Climbing experience [yrs] | CG | 9 | 1.54 | 1.49 | 0.30 | 5.00 |
| | ST | 9 | 2.94 | 1.59 | 1.00 | 5.00 |
| | WT | 8 | 2.63 | 2.37 | 1.00 | 8.00 |
| | Total | 26 | 2.36 | 1.87 | 0.30 | 8.00 |
| Training [h/wk] | CG | 9 | 2.56 | 1.13 | 0.50 | 4.00 |
| | ST | 9 | 2.39 | 0.99 | 1.00 | 4.00 |
| | WT | 8 | 2.56 | 0.56 | 2.00 | 3.50 |
| | Total | 26 | 2.50 | 0.91 | 0.50 | 4.00 |
| RP level [IRCRA] | CG | 9 | 15.00 | 1.54 | 13.50 | 17.00 |
| | ST | 9 | 16.06 | 1.70 | 13.50 | 18.00 |
| | WT | 8 | 16.38 | 1.41 | 15.00 | 18.00 |
| | Total | 26 | 15.79 | 1.61 | 13.50 | 18.00 |
| OS level [IRCRA] | CG | 9 | 13.89 | 1.29 | 12.00 | 16.00 |
| | ST | 9 | 14.61 | 1.52 | 12.00 | 16.00 |
| | WT | 8 | 14.69 | 0.96 | 13.50 | 16.00 |
| | Total | 26 | 14.39 | 1.29 | 12.00 | 16.00 |

IRCRA–International Rock Climbing Research Association reporting scale, RP performance–red point, OS–on sight performance

research as outlined in the 5th Declaration of Helsinki. Participants were informed (both in writing and orally) about all testing and training procedures and gave written informed consent to participate prior to entering the study. The data were accessed for research purposes on November 30th 2022. Authors had no access to information that could identify individual participants after data collection.

## Procedures

**Testing procedures.** A diagram of all testing procedures is presented in Fig 2, and an overview of all tests and corresponding measures can be found in the supplementary material (S1 Table). During the familiarization session subjects completed a questionnaire on climbing experience, climbing routine, health and physical well-being, as well as on-sight (OS) and redpoint (RP) levels. They were also asked to list any other sports they were regularly engaged in, apart from climbing. Furthermore, their height and weight were measured, using a metric stadiometer, attached to a wall, and a bioimpedance scale, respectively (Tanita MC 780MA S, Tokyo, Japan). Finally, subjects were familiarised with the four strength tests.

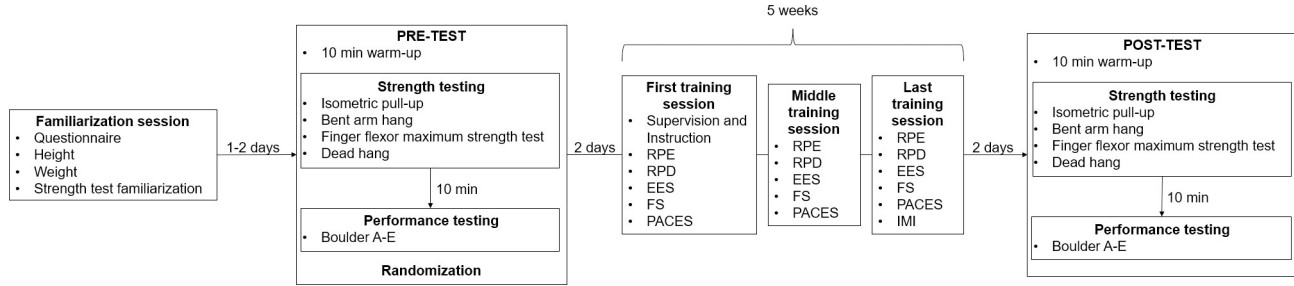

**Fig 2. Testing procedures.**

Pre- and post-testing started with a 10-minute standardized warm-up programme consisting of 30 jumping jacks, 10 forward and backward arm and shoulder circles, two pull ups on a 32 mm deep and 12.7 cm wide jug, and a short fingerboard warmup–five-second hangs each from the jug, the 45 mm and the 20 mm edge of the Beastmaker 1000 series fingerboard (Beastmaker LTD Carnforth, Lancashire, United Kingdom) with a five-second rest in between each repetition.

*Strength testing.* All four strength tests were chosen according to the findings of Langer et al. [43], proving all applied testing methods as reliable and valid measurements of climbing specific strength.

After the warm-up the subjects started with the upper-limb strength testing. They performed an isometric pull-up test on 32 mm deep and 12.7 cm wide jugs attached to two custom-made force plates (one for each arm), including a vertical force cell with 200 Hz resolution (Ergotest Innovation A/S, Porsgrunn, Norway) and were attached to a main board, held by a free-standing frame (Fig 3A). For the isometric pull-up test a metal bar was fixed to the lower part of this frame. Subjects wore a harness and were attached to this metal bar by two carabiners and an 18 mm wide nylon daisy chain (Diamond Equipment, Ltd, Salt Lake City, Utah, USA), stopping their movement at an elbow angle of 90 degrees. Subjects were asked to perform a pull-up and to keep pulling as hard as possible for five seconds after the upward movement was stopped (Fig 3A). The test was repeated three times with a one-minute rest in between each trial. The highest average force (N) over a time interval of three seconds was noted for each trial and the average from all three trials relative to body-weight was used for the analysis.

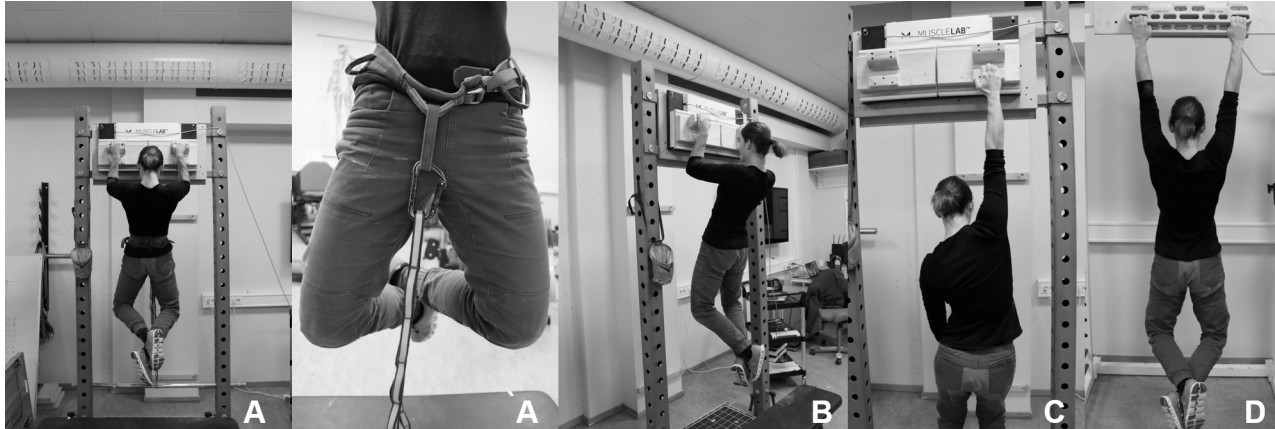

**Fig 3. Testing set ups.** A–Isometric pull-up, B–Bent arm hang, C–Finger strength measurement, D–Dead hang.

Following a three-minute rest, upper-limb muscular endurance was assessed using the bent arm hang test. Subjects were asked to hang from the same jugs with their elbows at a 90-degree angle for as long as possible (Fig 3B). The time was stopped as soon as their eyes dropped below the wooden board the jugs were attached to. Time until failure measured in seconds was used in the analysis. Due to its high intensity and fatigue, this test was conducted only once.

After a five-minute rest, subjects then performed the finger strength test. They were asked to apply as much of their body weight as possible to a 19 mm deep and 12.7 cm wide campus rung (Metolius Climbing, Bend, Oregon, USA), attached to the same measurement device, for five seconds with extended arms (Fig 3C). Subjects were not allowed to use the thumb and were asked to maintain an open or half crimp grip, depending on their preferences. They were also asked to use the same grip for pre- and post-testing. Three trials were performed alternately for the right and left hand with a one-minute rest between the trials. The highest average force (N) over a three-second time interval for each hand was determined from the force curves and the average from all six trials, relative to body-weight, was used for the analysis.

After another three-minute rest, the subjects performed the dead hang test to assess finger muscular endurance. In this test, the subjects were asked to hang for as long as possible from the 45 mm deep four-finger pockets with straight arms (Fig 3D). Again, they were not allowed to use their thumbs and were allowed to choose between a half-crimp or an open grip. Like the bent arm hang test, this test was only performed once and time until failure measured in seconds was used in the analysis.

*Performance testing.* After the strength tests, the subjects entered bouldering room (approximately 10-minute transition time) where they completed performance testing comprising a total of five bouldering routes. Two routes (A and B), both graded 6a on the Fontainebleau scale, with nine and six hand-holds respectively, were set up on a 12 x 12 Kilter board (Kilter, LLC, Boulder, Colorado, USA) with kickboard and adjustable angle at 15˚ inclination (S1A and S1B Fig). The other three routes (C, D, E) had seven, five, and five hand-holds respectively and were set up on a 4.2 m high boulder wall with an inclination of 20˚ (6a+), 10˚ (6b) and -5˚ (slab– 6a+) respectively (S1C–S1E Fig). The order of the routes was standardized and subjects were allowed four attempts on each boulder with a one-minute rest between attempts and a two-minute rest between the different boulders. The one-minute rest was allowed to be shorter if the subject fell on the first move. The hand-holds were numbered on each boulder and the highest hand-hold reached during the best attempt was noted. The sum of the highest holds reached on all routes (maximum 32) was used for the analysis. If a subject touched, but could not hold on to a hold, half a point was awarded. If a subject was able to complete a boulder, the number of attempts to reach the top hold was noted. If a subject was not able to complete a boulder within the four given tries, five attempts were noted. The total number of attempts for all boulders (maximum 25 –no top on any of the boulders) was used for the analysis. If a subject was unable to complete a boulder, the number of attempts to reach the top hold was noted as five.

In addition, subjects were filmed during their attempts on boulder B and C in order to rate climbing technique. While climbing technique cannot be generalized due to the high complexity and variety of climbing movements, researchers have identified some key components of "a good climbing technique" which have been translated into a validated performance assessment tool by Taylor et al. [44]. A shortened version of this evaluation tool by Taylor et al. [44] was used to assess accuracy, balance and fluidity, sequencing and exploration, skill repertoire, arm posture, and movement initiation (S2 Table). The best attempt on each boulder was rated according by a total of five independent experts. The experts were blinded to the group and time of the video and all faces were removed from the videos to avoid facial recognition. Rater one, two, and tree analysed the same videos, while rater four and five each assessed half of the

remaining videos. Ten of the videos given to all raters were identical in order to assess inter-rater agreement. To evaluate the intra-rater reliability, each rater was tasked with re-assessing a specific number of videos. Raters one, two and three were given four videos to reassess along with the rest of the video set. Rater four and five re-evaluated ten videos after a four-week period.

*Emotional and motivational measurements.* After the first, fifth (mid), and last training session, in-person questionnaires were conducted to assess the subjectss' training experience. Due to the fact that the subjects mainly trained independently, exercise intensity was not externally supervised. Rate of perceived exertion (RPE), which has been shown to be a reliable method of quantifying resistance training intensity [45, 46], was assessed to evaluate training intensity. Perceived training intensity and exercise discomfort were assessed using the rate of perceived exertion (RPE) and the rate of perceived discomfort (RPD) scales as introduced and validated by Steele et al. [47]. Exercise enjoyment was assessed through the exercise enjoyment scale (EES) validated by Stanley et al. [48], and the eight-point physical activity enjoyment scale (PACES) validated by Radeke [49]. Additionally, training pleasure was assessed, using the exercise feeling scale (FS) validated by Hardy and Rejeski [50]. All scales were explained to the participants in advance. The structure of the scales used is presented in Table 2.

After completing the five-week training intervention, subjects from ST and WT answered a questionnaire about their motivation to participate in a similar, self-organized, regular training program in the future. The intrinsic motivation inventory (IMI) introduced and validated by McAuley et al. [51] was adjusted to include a total of 24 items with four subscales (interest/enjoyment, effort/importance, pressure/tension, and value/usefulness) in a randomized order. The scale was explained to the subjects in advance and each question was rated on a unipolar 7-point Likert scale (1 = not at all true, 7 = very true) (S3 Table).

**Training program.** This program of both intervention groups lasted for approximately 30 minutes. The warm-up was equal for both groups and consisted of thirty jumping jacks, arm circling, shoulder circling, a short finger warm-up and five easy boulders (7–9 IRCRA).

**Table 2. Scales used to assess exercise enjoyment, discomfort and pleasure.**

| Scale | Lowest score | | Highest score |
|---|---|---|---|
| RPE | **0**<br>No exertion | | **10**<br>Maximal exertion |
| RPD | **0**<br>No discomfort | | **10**<br>Maximal discomfort |
| FS | **-5**<br>Very displeased | | **5**<br>Very pleased |
| EES | **1**<br>No enjoyment at all | | **7**<br>Extremely much enjoyment |
| PACES | **1** | | 7 |
| | 1. | I hate it | I enjoy it |
| | 2. | I feel boredom | I feel interested |
| | 3. | I dislike it | I like it |
| | 4. | It is very uncomfortable | It is very comfortable |
| | 5. | It is not fun at all | It is very fun |
| | 6. | It is very unpleasant | It is very pleasant |
| | 7. | I would rather be doing something else | There is nothing else I would rather be doing |
| | 8. | This activity is not attractive to me | This activity is attractive to me |

RPE–rate of perceived exertion, RPD–rate of perceived discomfort, FS–feeling scale (training pleasure), EES–exercise enjoyment scale, PACES–physical activity enjoyment scale

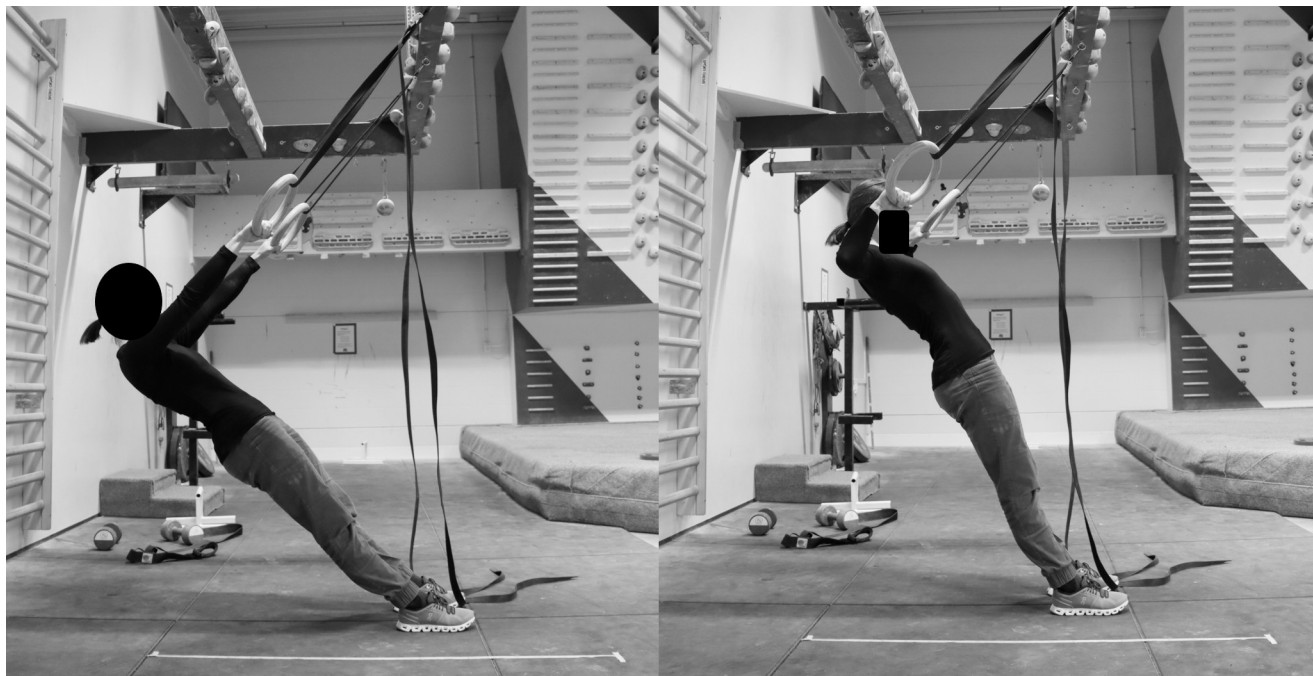

**Fig 4. Ring training exercise with numbered tape on the floor to control the intensity.**

*Off-the-wall group*. After the warm up, subjects from the ST performed ring rows (low pull-ups) on gymnastic rings with elbows pointing outwards. The rings were adjusted to chest height and participants did six sets of 25 pull ups with a one-minute rest between each set, according to the suggestions by Schmidtbleicher [38]. To ensure that the feet were placed on the same sport during a session, and to simplify intensity adjustments, a numbered tape was stuck to the floor next to the rings. The participants were asked to adjust the intensity for the following session by moving their feet further back (lower intensity) or forward (higher intensity) so that they were not able to easily do more than 25 repetitions per set but also did not have to stop after less than 20 repetitions to ensure intensities corresponding local muscular endurance training [38] (Fig 4).

Subsequently, participants proceeded to the fingerboard. They were instructed to perform one of the trainings marked as Level 1–4 shown in Table 3 using the 45 mm deep four-finger pockets on the beastmaker 1000 series fingerboard. All subjects started at Level 1. Over the course of the intervention, they proceeded to a next higher level if they were able to easily do the previous. In case they were not

able to complete the full program of a higher level, they were instructed to go back by one level but to add one set. For intermittent finger hangs, Winkler et al. [52] found that a hang to rest ratio of 7:2 seconds is most representative of the climbing movement. For this study the

**Table 3. Training levels on the fingerboard.**

| Level | Sets | 1 repetition | Repetitions per set | rest between sets |
|---|---|---|---|---|
| 1 | 3 | 10s hanging 30s rest | 6 | 1 min |
| 2 | 4 | 10s hanging 20s rest | 6 | 1 min |
| 3 | 5 | 10s hanging 10s rest | 6 | 1 min |
| 4 | 5 | 10s hanging 5s rest | 6 | 1 min |

hang-time was slightly increased, while rest times were strongly increased in order to adjust the intensity for individuals without previous fingerboard training experience, and to reduce risk of injury [11, 14].

*On-the-wall group.* Participants in the WT group were instructed to attempt four out of ten boulder problems of increasing difficulty on the Kilterboard after the warm up. They were allowed a maximum of four attempts in a five-minute window per boulder. For the following session, they omitted the easiest of the four and included one new problem. If they were struggling with all four bolder problems, they could repeat the same problems in the next session. This design was used to allow progression in difficulty while accommodating the varying performance levels of the participants. The 12 x 12 Kilter Wall with kickboard was set at a 15-degree angle during training.

Subjects from both groups were supervised and instructed individually on how to continue the training during their first training session. They then followed the program according to their own climbing schedule for five weeks. This meant for them to perform the training at the start of a bouldering or climbing session at least twice a week. They were allowed to perform the training more than twice a week but were instructed to not do the training on two consecutive days. Furthermore, they were instructed to change the intensity of the exercises for the next training session according to their subjective perception during the last session. The number of training sessions each participant completed, as well as the total time spent climbing and/or bouldering during the five-week intervention is presented in the supplementary material (S4 Table).

The CG performed the same warm-up at the start of their bouldering or climbing sessions but apart from that stuck to their usual climbing routine.

In addition, all subjects, including the CG, were asked to note any kind of sportive activity and training during the five weeks of the study in a training diary (S5 Table). This included exercise type, exercise duration and exertion on a scale from zero (not hard at all) to ten (extremely exhausting). The subjects in the ST and WT were additionally asked to note the intensity of the prescribed training programme. This included the position of the feet during the pull-ups, the level selected, as well as the number of sets performed on the fingerboard, and the boulders they attempted on the Kilterboard.

## Statistical analysis

JASP statistical software (JASP (Version 0.17.1.0) Computer software) was used for statistical analyses. Bayesian statistics were implemented as they are not based on large samples [53]. Further, Bayesian posterior distributions provide a better overview of the probability of all possible parameter values given the actual data, while frequentist analysis "only provides the probability that summaries of simulated data from a hypothetical value of the parameter would be more extreme than the summary of the actual data" [54]. Moreover, Bayesian statistics enable the presentation of evidence supporting a hypothesis, unlike frequentist hypothesis testing, which can only reject or fail to reject a specific hypothesis. This might serve to provide results with less bias toward rejected null hypotheses, and more precise interpretation of the data [54]. Due to the fact that Bayesian statistics, additionally, do not require the selection of appropriate error terms for F-ratios or corrections for multiple comparisons, Kruschke and Liddell [54] especially emphasize the value of Bayesian statistics for RCT studies.

Khant and Rayner [55] proved an analysis of variance (ANOVA) to be suitable for small sample sizes, even with non-normally distributed data. Differences between groups, subjects, and pre- and post-tests were thus calculated using multivariate and univariate Bayesian ANOVAs.

Test results are presented as means with standard deviations. Statistical results are presented including bayes factors, and posterior odds. Bayes factors were rated according to Goss-Sampson et al. [56].

**Performance and strength tests.** Differences between groups regarding subject characteristics were calculated using Bayesian one-way ANOVAs. The same was done for pre-test strength and performance measurements.

A Bayesian three-factor repeated measures MANCOVAs was conducted to explore potential differences across group, time, and performance tests (highest hold reached, number of attempts, expert ratings), in order to examine RH1 and RH3.

Another Bayesian three-factor repeated measures MANCOVAs was conducted to explore potential differences across group, time, and strength tests (bent arm hang, dead hang, finger strength, arm strength), in order to examine RH2.

As substantial differences between subjects were found for the hours spent climbing and/or bouldering during the five-week intervention, the hours climbed were introduced into the calculations as a covariate. In order to allow a comparison between the tests, all pre- and post-test data was Z-transformed and the total number of attempts were inverted.

Inter- and intra-rater agreement for expert ratings were calculated using Bayesian Correlation. Pearson's r was analyzed according to Schober et al. [57] ($r < .10$ –negligible correlation, $0.10 \leq r \geq 0.39$ –weak correlation, $0.40 \leq r \geq 0.69$ –moderate correlation, $0.70 \leq r \geq 0.89$ –strong correlation, $0.90 \leq r$–very strong correlation). Respective Bayes factors were also considered.

In addition to the qualitative analysis, individual responses were looked at, as suggested by Hecksteden et al. [58].

**Emotional and motivational measurements.** In order to examine RH 4, Bayesian two factor repeated measures ANOVAs were performed to analyse time, group, and time*group interaction effects for RPE, and RPD.

In order to examine RH5, Bayesian two factor repeated measures ANOVAs were performed to analyse time, group, and time*group interaction effects for FS, EES, and PACES. The mean of the sum of all eight questions was used for PACES analysis, as suggested by Kendzierski and DeCarlo [59]. Post-hoc testing included Bayesian t-tests controlled for multiplicity.

In addition, between group differences regarding IMI after the five-week intervention were calculated using a Bayesian Mann-Whitney U-test. Negative items were inverted prior to analysis.

Questionnaire consistency of the PACES scale and the four IMI subscales was analysed through Bayesian unidimensional reliability testing. MacDonald's ω was used, as suggested by Trizano-Hermosilla and Alvarado [60], and interpreted as follows: $ω < 0.6$ –unacceptable, $0.6 \geq ω < 0.7$ –questionable, $0.7 \geq ω < 0.8$ acceptable, $0.8 \geq ω < 0.9$ good, $ω \geq 0.9$ –excellent [61].

The consistency of subjects' responses across scales was assessed by examining the correlation between comparable items from different scales (PACES 1-EES-IMI 1, PACES 2-IMI 23, PACES 4-RPD, PACES 5-IMI 9) employing Bayesian Correlation. Pearson's r was interpreted as described for intra- and inter-rater agreement of the expert ratings.

The number of hours spent on other sports during the time of the intervention by each individual was extracted from the training diaries and analyzed quantitatively.

## Results

Subject characteristics by group are presented in Table 1. The results from the Bayesian one-way ANOVAs revealed only anecdotal evidence for the model including "group" to explain

**Table 4. Bayesian output of the one-way ANOVAs regarding subject characteristics and pre-test measurements.**

| Factor | Model | P(M) | P(M\|data) | BF$_M$ | BF$_{10}$ | error % |
|---|---|---|---|---|---|---|
| Age | Group | 0.500 | 0.195 | 0.242 | 0.242 | 0.037 |
| Height | Group | 0.500 | 0.231 | 0.300 | 0.300 | 0.024 |
| Weight | Group | 0.500 | 0.253 | 0.339 | 0.339 | 0.026 |
| Climbing experience [yrs] | Group | 0.500 | 0.358 | 0.559 | 0.559 | 0.015 |
| Training [h/wk] | Group | 0.500 | 0.197 | 0.246 | 0.246 | 0.037 |
| Best RP | Group | 0.500 | 0.414 | 0.706 | 0.706 | 0.018 |
| Best OS | Group | 0.500 | 0.304 | 0.346 | 0.346 | 0.013 |
| Upper-limb strength | Group | 0.500 | 0.761 | 3.188 | 3.188 | 0.006 |
| Bent arm hang | Group | 0.500 | 0.345 | 0.528 | 0.528 | 0.015 |
| Finger strength | Group | 0.500 | 0.336 | 0.507 | 0.507 | 0.014 |
| Dead hang | Group | 0.500 | 0.292 | 0.413 | 0.413 | 0.026 |
| Total number of attempts | Group | 0.500 | 0.343 | 0.522 | 0.522 | 0.015 |
| Sum of highest holds reached | Group | 0.500 | 0.356 | 0.553 | 0.553 | 0.015 |
| Expert ratings | Group | 0.500 | 0.360 | 0.563 | 0.563 | 0.018 |

P(M)–prior model probability, P(M|data)–posterior model probability, BF$_M$–posterior model odds, BF$_{10}$ –Bayes factor (evidence for the alternative hypothesis relative to the null hypothesis/null model), error %–error of the Gaussian quadrature integration routine used for the computation of the Bayes factor

differences in subject characteristics, and performance measurements at pre-test, when compared to the null model. Similar results were found for strength measurements. Anecdotal evidence for differences between groups was found for finger strength, and upper limb and finger muscular endurance. Moderate evidence for group differences at pre-test were found for upper-limb strength (Table 4).

### Performance tests

The development of all performance measurements is presented in Fig 5A–5C. Means and standard deviations are presented in the supplementary material (S6 Table). The overall number of attempts decreased in all groups from pre- to post-test. CG consistently exhibited higher values compared to both intervention groups. Similar trends were observed in the sum of the highest holds reached across all five boulders, with improvements seen in all groups from pre- to post-test and the control group generally displaying lower values than the intervention groups.

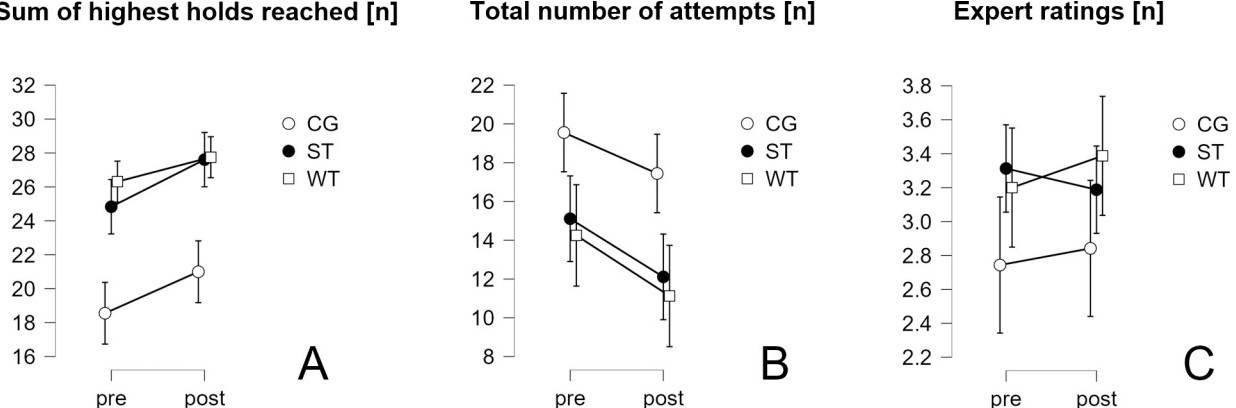

**Fig 5. Performance development over time and confidence intervals (CI95%) presented by group.** CG–control group, ST–off-the-wall training group, WT–on-the-wall training group.

**Table 5. Results from the Bayesian three-factor repeated measures MANCOVA (group, time, performance test)–showing best 5 out of 38 models.**

| Models | P(M) | P(M\|data) | BF$_M$ | BF$_{10}$ | error % |
|---|---|---|---|---|---|
| Null model (incl. subject and random slopes) | 0.026 | 0.038 | 1.459 | 1.000 | |
| Group + climbing [h] | 0.026 | 0.264 | 13.267 | 6.959 | 18.606 |
| climbing [h] | 0.026 | 0.233 | 11.221 | 6.136 | 2.683 |
| test + climbing [h] | 0.026 | 0.097 | 3.964 | 2.552 | 13.893 |
| test + Group + climbing [h] | 0.026 | 0.066 | 2.595 | 1.728 | 4.649 |
| time + climbing [h] | 0.026 | 0.054 | 2.101 | 1.417 | 2.794 |

Climbing [h]–the total time spent climbing and bouldering during the time of the intervention (covariate introduced into the calculation), P(M)–prior model probability, P(M\|data)–posterior model probability, BF$_M$–posterior model odds, BF$_{10}$ –Bayes factor (evidence for the alternative hypothesis relative to the null hypothesis/null model), error %–error of the Gaussian quadrature integration routine used for the computation of the Bayes factor

Expert ratings were assessed for twenty-one participants. Three subjects from the CG, and one subject from both the WT and the ST group were unable to execute the start of boulder B or C during both the pre-test and post-test. Consequently, the experts were unable to provide ratings for these individuals.

Inter-rater for the evaluation of the climbing technique revealed a moderate to strong inter-rater correlation (r = .453 –.741). Bayes factors supported these findings with anecdotal to moderate evidence for a positive correlation between most raters. Anecdotal evidence against a positive correlation was found for one rater compared to two others (S7 Table). Intra-rater agreement ranged from, moderate to strong for four of the five raters (r = .655 –.857). This was accompanied by large Bayes factors, indicating decisive evidence for these findings. For the remaining rater (rater 1), only a weak correlation (r = .185), accompanied by anecdotal evidence for the null hypothesis, could be found (S8 Table). All analyses were thus performed without the participation of rater one.

As shown in Fig 5C, both CG and WT expert ratings slightly improved from pre- to post-test, with WT demonstrating generally higher values. ST, however, demonstrated a slight decrease.

The Bayesian three-factor MANCOVA for group, time, and performance test under the assumption of equal a priori likelihood of the models, indicated that the data are most likely under a model including group and hours climbed during the intervention moderate evidence. Moderate evidence was also found for a model including only hours climbed during the intervention (Table 5).

In accordance with RH1, post-hoc comparisons for group revealed moderate evidence for a difference between CG and ST, and decisive evidence for a difference between CG and WT (Table 6). In addition, we found anecdotal evidence against differences between ST and WT (Table 6), and thus against RH3.

**Table 6. Bayesian repeated measures MANCOVA of performance measurements post hoc comparisons–group.**

| Group | | Prior Odds | Posterior Odds | BF$_{10}$ | error % |
|---|---|---|---|---|---|
| CG | ST | 0.587 | 4.991 | 8.496 | 4.314+10–7 |
| | WT | 0.587 | 4791.363 | 8156.886 | 4.178*10–6 |
| ST | WT | 0.587 | 0.309 | 0.526 | 0.016 |

CG–control group, ST–off-the-wall training group, WT–on-the wall training group, Prior Odds–The odds of the outcome before the evidence is considered, Posterior Odds–The odds of the outcome considering the evidence, BF$_{10}$ –Bayes factor (evidence for the alternative hypothesis relative to the null hypothesis/null model), error %–error of the Gaussian quadrature integration routine used for the computation of the Bayes factor

## Strength tests

The development of all strength measurements is presented in Fig 6A–6D. Means and standard deviations are presented in the supplementary material (S6 Table). Upper-limb strength improved across all groups from pre- to post-test, with the control group experiencing the greatest increase. Similar increases across all groups were found for finger strength. Upper-limb muscular endurance improved in all groups, with the control group generally displaying lower values compared to both intervention groups. Both ST and CG demonstrated increases in muscular endurance of the finger flexors, with ST showing larger gains. Meanwhile, WT slightly decreased from pre- to post-test.

The Bayesian three-factor repeated measures MANCOVA under the assumption of equal a priori likelihood of the models revealed models including group or hours climbed during the intervention to explain the data slightly better than the null model (anecdotal evidence $BF_{10}$ =

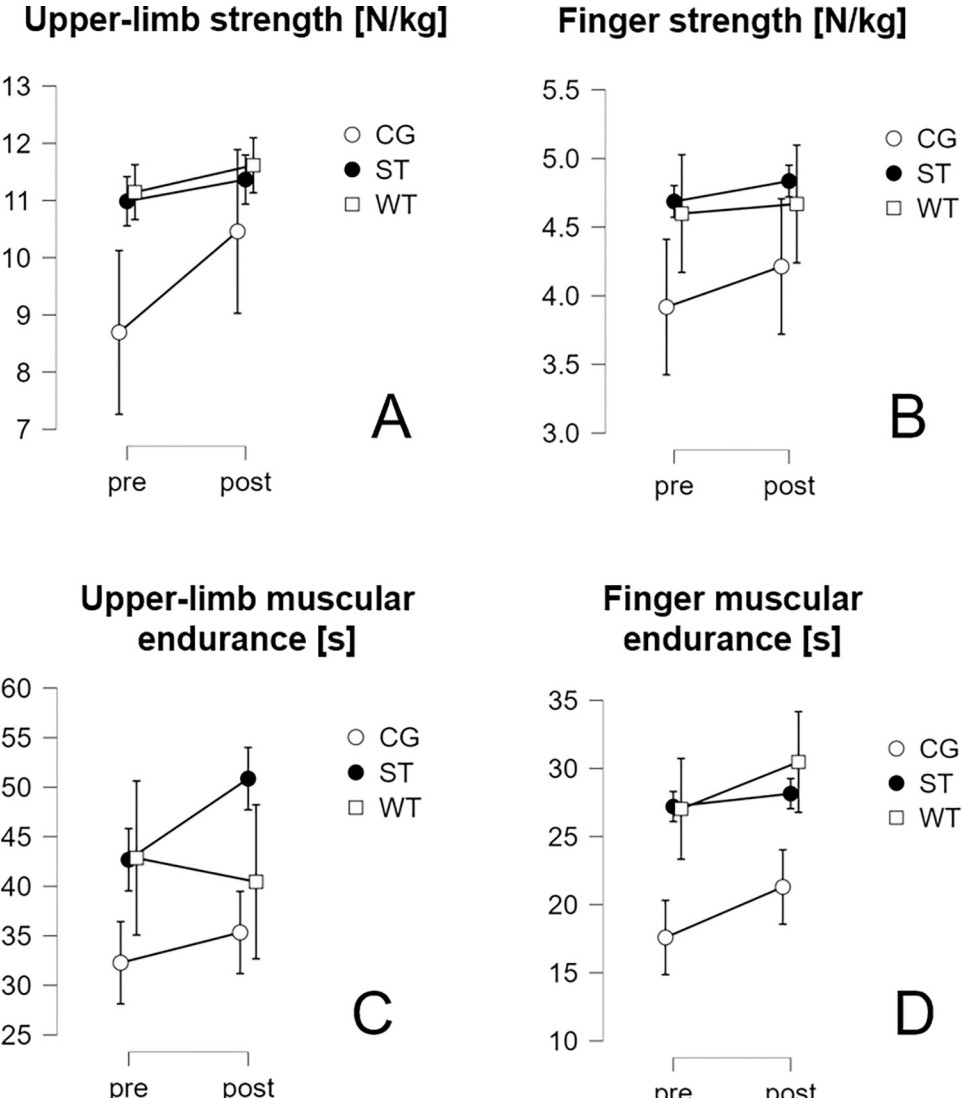

**Fig 6. Strength development over time and confidence intervals (CI95%) presented by group.** CG–control group, ST–off-the-wall training group, WT–on-the-wall training group.

**Table 7. Results from the Bayesian three-factor repeated measures MANCOVA (group, time, strength test)–showing best 5 out of 38 models.**

| Models | P(M) | P(M\|data) | BF$_M$ | BF$_{10}$ | error % |
|---|---|---|---|---|---|
| Null model (incl. subject and random slopes) | 0.026 | 0.180 | 8.115 | 1.000 | |
| Group | 0.026 | 0.209 | 9.749 | 1.159 | 9.340 |
| climbing [h] | 0.026 | 0.193 | 8.846 | 1.073 | 1.590 |
| Group + climbing [h] | 0.026 | 0.170 | 7.579 | 0.945 | 4.294 |
| time + climbing [h] | 0.026 | 0.044 | 1.694 | 0.243 | 14.962 |
| time | 0.026 | 0.036 | 1.369 | 0.198 | 1.733 |

Climbing [h]–the total time spent climbing and bouldering during the time of the intervention (covariate introduced into the calculation) P(M)–prior model probability, P(M\|data)–posterior model probability, BF$_M$–posterior model odds, BF$_{10}$ –Bayes factor (evidence for the alternative hypothesis relative to the null hypothesis/null model), error %–error of the Gaussian quadrature integration routine used for the computation of the Bayes factor

1.159, BF$_{10}$ = 1.073) (Table 7). Due to the fact that only anecdotal evidence was found for other models than the null model, no further analyses were conducted, as suggested by et al. Goss-Sampson et al. [56]. We can thus state that only anecdotal evidence in favour of RH2 was found.

## Individual responses

Due to the small number of subjects in the study, individual strength and performance development from pre- to post-test was plotted for each subjects, as suggested by Hecksteden et al. [58] (S2A–S2I and S3A–S3D Figs; and S9 Table).

This revealed that the total number of attempts most strongly improved in individuals of WT. The sum of the highest holds reached increased similarly in most individuals. Expert ratings both increased and decreased in all groups. Looking more closely at the different technical factors, only small interindividual differences could be observed for balance and accuracy. Especially individuals of CG seemed to improve in both these factors, while individuals of WT also improved more in accuracy compared to individuals of ST. Technique ratings stayed constant from pre- to post-test for most individuals, except for some slight decreases and increases in individuals of all groups. Movement initiation through the hips improved in WT and CG but decreased for most individuals of ST. Sequencing both improved and decreased for individuals of all groups. Arm posture mainly improved in individuals form the ST group and decreased for CG and WT.

Regarding strength measurements, individuals from all groups improved in finger muscular endurance. Compared to the other groups, most individuals improved in ST, also showing the strongest improvements, compared to the other groups. Individuals of CG especially improved more strongly in upper-limb strength, compared to the other groups. Especially individuals of CG improved in upper-limb muscular endurance, while individuals of the ST group showed only very small improvements. Some individuals from WT improved strongly, while the performance of others stayed the same or decreased very slightly.

Individuals of all three groups improved in finger strength, while individuals of CG showed the highest improvements followed by ST and then WT. Quantitative analysis of the total number of hours spent on other sports during the time of the intervention did not show differences between groups (S5 Fig). One subject from WT was excluded from this analysis as they did not use the standardized training diary and did not provide any data on other sportive activities apart from the prescribed training.

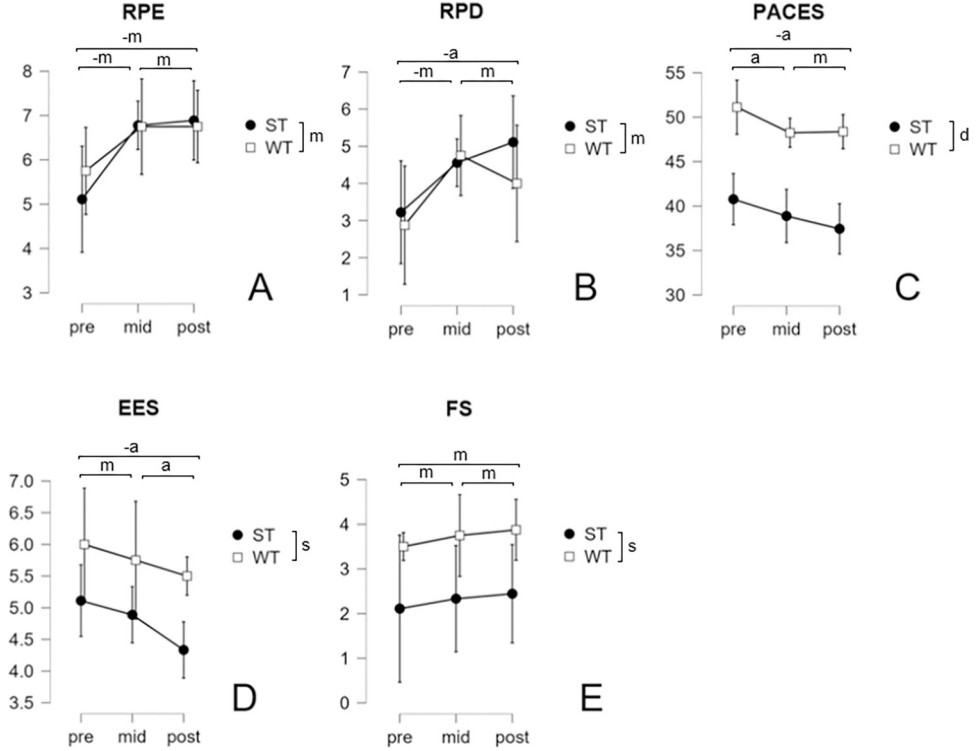

**Fig 7. Development of emotional measurements over time and confidence intervals (CI95%) presented by group.**
RPE–Rate of perceived exertion, RPD–rate of perceived discomfort, PACES–physical activity enjoyment scale, EES–
exercise enjoyment scale, FS–feeling scale, CG–control group, ST–off-the-wall training group, WT–on-the-wall
training group, a/m/s/d–anecdotal/moderate/strong/decisive evidence regarding group and time differences in favour
of RH4/RH5 or, in the case of a negative prefix, in opposition to RH4/RH5 (Regarding RPE and RPD, RH4 expects no
group differences).

### Emotional and motivational measurements

The development of all emotional measurements is presented in Fig 7A–7E. Means and standard deviations are presented in the supplementary material (S10 Table).

The posterior mean for PACES revealed good consistency. The removal of item four improved reliability slightly. Deleting any of the other items leads to a decrease in reliability. Therefore, values of all items were analyzed collectively (sum of all items). Posterior Plots are presented in the supplementary material (S4A Fig).

For the IMI subscale interest/enjoyment, analysis revealed questionable consistency. Deleting item 2 or 23, however, led to an increase to acceptable consistency with a higher increase after the deletion of item 2. The deletion of any other item led to a decrease in reliability. IMI-interest/enjoyment values were thus analyzed without item 2. Acceptable consistency was found for the value/usefulness subscale. The removal of any item led to a slight decrease in consistency. Unacceptable consistency was found for the effort/importance, and pressure/tension subscales. These two subscales were thus not included in further analyses. All Posterior Plots are presented in the supplementary material (S4B–S4E Fig).

In addition, correlation analysis between PACES and EES ratings revealed strong correlation, supported by decisive evidence (r = .802, $BF_{10}$ = 292.220).

In addition, correlation analysis between single items of different scales revealed strong correlation between EES and PACES item 1, as well as between IMI item 1 and PACES item 1,

and between IMI item 1 and EES. These findings were, additionally, supported by very strong to decisive evidence through large Bayes factors.

Moderate negative correlation supported by moderate evidence was found for PACES item 2 and IMI item 23. Weak correlation with anecdotal evidence against a positive correlation was found for PACES item 4 and RPD. Moderate correlation and moderate evidence were found for PACES item 5 and IMI item 9. All values can be found in the supplementary material (S11 Table).

RPE was similar in both intervention groups. It increased from the first to the middle session but then stayed constant until the last session. RPD was similar for both groups during the initial session and rose for both groups by the middle session. It decreased for WT and increased for ST by the last session. PACES values were much higher in WT compared to ST. They slightly decreased from the initial to the middle session in both groups and then stayed constant for WT, while continuing to slightly decrease in ST. ESS values were also higher in the WT group compared to the ST. They slightly decreased in both groups from the initial- to the middle session and from the middle session to the last session. ESS values were higher in the WT group compared to the ST. They slightly increased in both groups from the initial- to the middle session and from the middle session to the last session. IMI measurements after the training intervention were slightly higher for WT compared to ST across all factors (interest/enjoyment, effort/importance, pressure/tension, value/usefulness).

Bayesian repeated measures ANOVA for RPE under the assumption of equal a priori likelihood of the models revealed very strong evidence for a model including time. Additionally, strong evidence was shown for a model including time and group (Table 8). Post-hoc comparisons revealed moderate evidence in favour of a difference between pre- and mid-, as well as between pre- and post-measurements in opposition to RH4. However, moderate evidence was found against a difference between mid- and post-measurements (Table 9). Furthermore, moderate evidence against a difference between ST and WT, and thus in favour of RH4, was found (Table 10).

Bayesian repeated measures ANOVA for RPD under the assumption of equal a priori likelihood of the models indicated moderate evidence for two models, including time, and both time and group, respectively (Table 8). Post-hoc comparisons for time revealed moderate evidence for a difference between pre- and mid-, as well as anecdotal evidence for a difference between pre- and post-measurements in opposition to RH4. However, moderate evidence against a difference between mid- and post-measurements was found (Table 9). In regards to group, moderate evidence against a difference between ST and WT, and thus in favour of RH4, was found (Table 10).

Bayesian repeated measures ANOVA for and EES under the assumption of equal a priori likelihood of the models indicated moderate evidence for two models, including time, and both time and group, respectively (Table 8). Post-hoc comparisons for time regarding EES revealed moderate evidence against a difference between pre- and mid-, and anecdotal evidence against a difference between mid- and post-measurements in favour of RH5. However, anecdotal evidence for a difference between pre- and post-measurements in opposition to RH5 were found.(Table 9). Regarding group, strong evidence for a difference between ST and WT, was found in favour of RH5(Table 10).

Bayesian repeated measures ANOVA for FS under the assumption of equal a priori likelihood of the models revealed moderate evidence for a model including group, compared to the null model ($BF_{10} = 4.918$) (Table 8). Post-hoc comparisons revealed moderate evidence against differences between all measurements and strong evidence for a difference between ST and WT in favour of RH5 (Tables 9 and 10).

**Table 8. Results from the Bayesian repeated measures ANOVAs for RPE, RPD, ESS, FS, and PACES measurements.**

| Models | P(M) | P(M\|data) | $BF_M$ | $BF_{10}$ | error % |
|---|---|---|---|---|---|
| **RPE** | | | | | |
| Null model (incl. subject and random slopes) | 0.200 | 0.011 | 0.043 | 1.000 | |
| time | 0.200 | 0.644 | 7.220 | 60.463 | 0.552 |
| time + Group | 0.200 | 0.257 | 1.386 | 24.185 | 2.408 |
| time + Group + time * Group | 0.200 | 0.084 | 0.369 | 7.936 | 1.193 |
| Group | 0.200 | 0.004 | 0.016 | 0.375 | 2.931 |
| **RPD** | | | | | |
| Null model (incl. subject and random slopes) | 0.200 | 0.066 | 0.282 | 1.000 | |
| time | 0.200 | 0.561 | 5.102 | 8.510 | 1.001 |
| time + Group | 0.200 | 0.247 | 1.309 | 3.744 | 1.242 |
| time + Group + time * Group | 0.200 | 0.099 | 0.440 | 1.504 | 2.605 |
| Group | 0.200 | 0.028 | 0.115 | 0.424 | 1.985 |
| **EES** | | | | | |
| Null model (incl. subject and random slopes) | 0.200 | 0.066 | 0.284 | 1.000 | |
| time + Group | 0.200 | 0.442 | 3.169 | 6.667 | 3.487 |
| Group | 0.200 | 0.288 | 1.619 | 4.345 | 0.568 |
| time + Group + time * Group | 0.200 | 0.113 | 0.508 | 1.701 | 1.238 |
| time | 0.200 | 0.091 | 0.399 | 1.369 | 0.710 |
| **FS** | | | | | |
| Null model (incl. subject and random slopes) | 0.200 | 0.137 | 0.633 | 1.000 | |
| Group | 0.200 | 0.677 | 8.381 | 4.954 | 1.902 |
| time + Group | 0.200 | 0.130 | 0.598 | 0.952 | 1.965 |
| time + Group + time * Group | 0.200 | 0.031 | 0.126 | 0.224 | 1.625 |
| time | 0.200 | 0.026 | 0.106 | 0.189 | 0.807 |
| **PACES** | | | | | |
| Null model (incl. subject and random slopes) | 0.200 | 0.003 | 0.011 | 1.000 | |
| time + Group | 0.200 | 0.559 | 5.077 | 199.338 | 2.342 |
| Group | 0.200 | 0.278 | 1.542 | 99.163 | 10.286 |
| time + Group + time * Group | 0.200 | 0.153 | 0.723 | 54.570 | 5.130 |
| time | 0.200 | 0.007 | 0.026 | 2.321 | 0.581 |

RPE–Rate of perceived exertion, RPD–rate of perceived discomfort, EES–exercise enjoyment scale, FS–feeling scale, PACES–physical activity enjoyment scale, P(M)–prior model probability, P(M|data)–posterior model probability, $BF_M$–posterior model odds, $BF_{10}$ –Bayes factor (evidence for the alternative hypothesis relative to the null hypothesis/null model), error %–error of the Gaussian quadrature integration routine used for the computation of the Bayes factor

Bayesian repeated measures ANOVA for PACES under the assumption of equal a priori likelihood of the models revealed a model including time and group as most predictive of the data (Table 8). While only anecdotal evidence supports a model only including time, the model including only group was supported by very strong evidence (Table 8). Post-hoc comparisons revealed anecdotal evidence against differences between pre- and mid-, and moderate evidence against differences between mid- and post-measurements in favour of RH5. Anecdotal evidence was found for differences between pre- and post-measurements in opposition to RH5 (Table 9). In addition, decisive evidence for a difference between ST and WT was found in favour of RH5 (Table 10).

Intrinsic training motivation for both intervention groups after the last training session is presented in Fig 8A–8D. While high values for interest/enjoyment, effort/importance, and value/usefulness for both groups were found, pressure/tension values were low in both ST and

**Table 9. Bayesian repeated measures ANOVA post hoc comparisons–time for RPE, RPD, EES, FS, and PACES.**

| | | Prior Odds | Posterior Odds | $BF_{10}$ | error % |
|---|---|---|---|---|---|
| **RPE** | | | | | |
| Pre | Mid | 0.587 | 4.780 | 8.138 | $5.624 \times 10^{-7}$ |
| | Post | 0.587 | 4.828 | 8.220 | $5.589 \times 10^{-7}$ |
| Mid | Post | 0.587 | 0.149 | 0.253 | 0.013 |
| **RPD** | | | | | |
| Pre | Mid | 0.587 | 6.390 | 10.878 | $5.329 \times 10^{-7}$ |
| | Post | 0.587 | 1.138 | 1.937 | $1.814 \times 10^{-6}$ |
| Mid | Post | 0.587 | 0.147 | 0.251 | 0.013 |
| **EES** | | | | | |
| Pre | Mid | 0.587 | 0.184 | 0.313 | 0.016 |
| | Post | 0.587 | 2.865 | 4.878 | $6.043 \times 10^{-7}$ |
| Mid | Post | 0.587 | 0.676 | 1.151 | 0.024 |
| **FS** | | | | | |
| Pre | Mid | 0.587 | 0.160 | 0.273 | 0.014 |
| | Post | 0.587 | 0.188 | 0.319 | 0.016 |
| Mid | Post | 0.587 | 0.153 | 0.260 | 0.014 |
| **PACES** | | | | | |
| Pre | Mid | 0.587 | 0.721 | 1.228 | 0.024 |
| | Post | 0.587 | 1.899 | 3.232 | $4.330 \times 10^{-7}$ |
| Mid | Post | 0.587 | 0.183 | 0.312 | 0.016 |

Prior- and Posterior odds have been corrected for multiple testing, Bayes factors are uncorrected.

RPE–Rate of perceived exertion, RPD–rate of perceived discomfort, EES–exercise enjoyment scale, FS–feeling scale, PACES–physical activity enjoyment scale, Prior Odds–The odds of the outcome before the evidence is considered, Posterior Odds–The odds of the outcome considering the evidence, $BF_{10}$–Bayes factor (evidence for the alternative hypothesis relative to the null hypothesis/null model), error %–error of the Gaussian quadrature integration routine used for the computation of the Bayes factor

WT. Pressure/tension values were similar in both groups. WT, however, showed slightly higher values in all other subscales. A Bayesian Mann-Whitney-U Test revealed moderate evidence for group differences regarding interest/enjoyment, and value/usefulness in favour of RH5 (Table 11).

We can thus report strong to decisive evidence in favour of RH5 regarding exercise enjoyment and training pleasure, and moderate evidence in favour of RH5 regarding intrinsic training motivation.

**Table 10. Bayesian repeated measures ANOVA post hoc comparisons–group for RPE, RPD, EES, FS, and PACES.**

| Measurement | Factors | | Prior Odds | Posterior Odds | $BF_{10}$ | error % |
|---|---|---|---|---|---|---|
| RPE | ST | WT | 0.587 | 0.176 | 0.300 | 0.008 |
| RPD | ST | WT | 0.587 | 0.212 | 0.361 | 0.008 |
| EES | ST | WT | 0.587 | 29.756 | 50.657 | $1.512 \times 10^{-7}$ |
| FS | ST | WT | 0.587 | 24.884 | 42.363 | $2.129 \times 10^{-}$ |
| PACES | ST | WT | 0.587 | 686194.075 | $1.168 \times 10^{+6}$ | $1.144 \times 10^{-11}$ |

Prior- and Posterior odds have been corrected for multiple testing, Bayes factors are uncorrected.

RPE–Rate of perceived exertion, RPD–rate of perceived discomfort, EES–exercise enjoyment scale, FS–feeling scale, PACES–physical activity enjoyment scale, WT–on-the-wall training group, ST–off-the-wall training group, Prior Odds–The odds of the outcome before the evidence is considered, Posterior Odds–The odds of the outcome considering the evidence, $BF_{10}$–Bayes factor (evidence for the alternative hypothesis relative to the null hypothesis/null model), error %–error of the Gaussian quadrature integration routine used for the computation of the Bayes factor

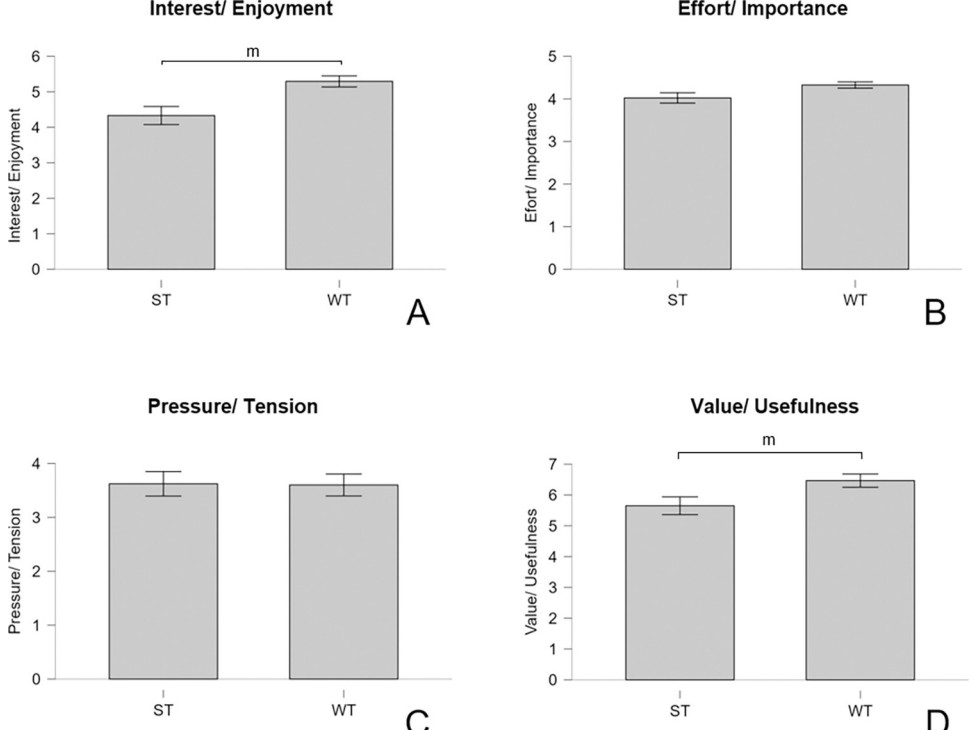

**Fig 8. Four sub-scales of the intrinsic motivation inventory presented by group at post-test, including confidence intervals (CI95%).** ST–off-the-wall training group, WT–on-the-wall training group, m–moderate evidence in favour of RH5.

All relevant raw data are provided in the supplementary material (S12 Table).

## Discussion

### Performance and strength

We hypothesised that both intervention groups would improve in bouldering performance but not the control group (RH1). Moderate, and decisive evidence was found for a higher increase in ST compared to CG and a higher increase in WT compared to CG, respectively.

In regards to climbing technique, larger training effects associated with on-the-wall training, compared to off-the-wall training were expected (RH3). Analysis of individual responses showed that expert ratings in CG and WT improved, while ST ratings slightly decreased. Looking more closely, in contrast to the other groups, ST expert ratings tended to decrease in precision of hand and foot placements and movement initiation. Arm posture, however, tended to improve over the course of the study. Individuals of WT, on the other hand, tended to improve

**Table 11. Results from the Bayesian Mann-Whitney U Test for IMI measurements.**

| Models | BF$_{10}$ | W | Rhat |
|---|---|---|---|
| Interest/ Enjoyment | 1.068 | 17.500 | 1.000 |
| Value/ Usefulness | 1.772 | 11.500 | 1.001 |

BF$_{10}$ –Bayes factor quantifying evidence for the alternative hypothesis relative to the null hypothesis, Rhat—Gelman-Rubin statistic

in movement initiation, and accuracy. This could point towards training on-the-wall to especially improve climbing flow, and precision. Furthermore, training off-the-wall without additional technique training might have led to a less accurate and less efficient way of climbing due to an increase in finger strength or an increase in trust into the fingers. These findings may also be explained by exercise specificity which was higher for WT compared to ST and has been shown to play an important role in training effects in climbing and in other sports [8, 18, 19]. Importantly, inter- and intra-rater reliability ranged between moderate and strong, which has to be kept in mind when interpreting these findings.

In regards to climbing-specific strength, larger training effects associated with off-the-wall training, compared to on-the-wall training were expected (RH2). Our results, however, showed no effect for time or group in regards to any of the measurements. Analysis of individual training responses showed only marginal differences between groups for upper-limb muscular endurance, and finger flexor and upper-limb strength. However, a slightly higher increase in dead hang holding times for ST were found when compared to individuals of the other groups, indicating a trend towards positive treatment effects for ST.

Together with the results from the individual-response analysis for performance measurements and the fact that no differences between groups were found for hours spent on other sports during the course of the intervention, our findings suggest an early onset of training effects. This is underlined by our findings indicating moderate evidence for an effect of both climbing time and group on performance measures. Potential explanations for the lack of measurable training effects might thus be attributed the short intervention period or insufficient training volume, hindering the occurrence of sufficient adaptions to the training. A study by Medernach et al. [21] found significant effects in upper-limb muscular endurance after only four weeks of muscular endurance training in advanced male climbers. Their training regimen, however, included 150 minutes of interval bouldering three times a week, adding up to a much higher training volume, compared to this study. Also, Hermans et al. [13, 15] found significant training effects on upper-limb muscular endurance after ten weeks of training, including two sessions a week, in intermediate climbers in two studies. This underscores the potential for the intervention period to have been insufficiently long and/or the weekly training volume being too low.

Analysis of strength and performance measures showed similar improvements for the control and the intervention groups across all tests. The analysis of the training diaries showed no differences between groups regarding the training volume in sports not related to the study. However, as shown by multiple studies, individuals with little experience and low initial strength levels experience smaller effort-to-benefit ratios compared to individuals with more experience and higher initial strength levels [62]. This could be a possible explanation for similar improvements in the control group compared to the intervention groups underlining the fact that lower-grade climbers, especially, already profit from regular climbing. This is in line with our findings indicating moderate evidence for an effect of the hours climbed during the intervention. Advanced climbers may, however, profit from additional training [7]. Since subjects were not under continuous supervision throughout the study, the possibility of incorrect exercise execution and insufficient training intensity, which have been identified as key factors for training efficacy [63], cannot be entirely dismissed for the intervention groups.

## Emotional and motivational measurements

As hypothesised, we found similar RPE values in both intervention groups (RH4). The fact that RPE was significantly lower in the first compared to the other training sessions in both WT and ST, could be explained by the fact that the first session served as an introduction

session to the training regimen. Both groups were, however, able to find an exertion level associated with hard physical work until the second measurement and kept this constant until the last session. While RPE is proven to be a valid tool for quantifying training loads (subjective strain) and training intensity distribution [64–66]. The subjective nature of RPE ratings, however, may limit their ability to accurately measure the true intensity of a resistance training [66]. Even though, subjects of both intervention groups were able to self-regulate and maintain training intensities over the course of the study, it cannot be evaluated whether training intensity corresponded to the recommended intensity for muscular endurance training. In addition, it was found that RPE ratings increased from the first to the second measurement, and then staying constant until the last session. It is thus unclear whether training intensities were low throughout the entire first half of the study or if they were low only in the first session.

RPD ratings were expected to be similar in both intervention groups (RH4). Results confirmed these expectations. However, the development of RPD over the course of the study shows a trend towards increasing discomfort in ST towards the end of the study. This could point towards the fact that training on the fingerboard, even with low intensities, can induce high discomfort in inexperienced, lower-level to advanced climbers and should thus only be incorporated into a training regimen in respect to individual preferences, and training comfort. However, there was a lack of consistency in responses related to exercise comfort (RPD and PACES 4), suggesting indistinct feelings in regards to training comfort. The same was found for boredom (IMI23 and PACES 2), and fun (IMI 9 and PACES 5).

Lastly, we expected higher exercise enjoyment, training pleasure, and intrinsic training motivation in WT compared to ST (RH5). It has been acknowledged that the sense of accomplishment derived from working hard is a key factor in determining the enjoyment of a workout [67]. Hence, it is not surprising that both training regimen were generally rated positively among the participants. Still, our results indicate higher exercise enjoyment for on-the-wall training, as hypothesized in advance. This was found for both PACES and EES ratings. Analysis of PACES showed good consistency. In addition, PACES item one was shown to strongly correlate with EES values, and IMI item one, supporting these findings. Furthermore, FS ratings were higher in WT compared to ST, indicating higher training pleasure.

These findings align with our expectations and could be explained by the fact that participants might have found the activity of bouldering itself satisfying rather than engaging in other types of training to improve bouldering performance. This may be explained by the fact that all recruited participants were climbers and were likely already intrinsically motivated to climb. Furthermore, on-the-wall training likely provided a more diverse training experience due to the availability of various boulder problems, compared to the resistance training protocol which included only two distinct exercises. According to the self-determination theory [26], autonomously selecting which boulder problems to work on might have been perceived as more motivating.

Nonetheless, it needs to be kept in mind that RPE, RPD, FS, EES and PACES measurements were only obtained in three out of ten training sessions, providing a restricted insight into participants complete experience of the training. However, intervening in every session could have been seen as disruptive to the participants and may have resulted in a decreased sense of autonomy in the training.

Furthermore, it should be noted, that while the interest/enjoyment subscale is considered to be the self-reporting measure of intrinsic motivation, the remaining subscales might account for outcomes of intrinsic motivation, rather than intrinsic motivation itself [68]. In addition, analysis of the subscales showed acceptable consistency for IMI subscales interest/ enjoyment (without items 2 or 23), and value/usefulness. Consistency for IMI subscales pressure/tension and effort/importance were, however, unacceptable.

The absence of a follow-up questionnaire to assess whether participants maintained their training alongside their regular bouldering routine does not allow to check for a possible intention-action gap [69]. Additionally, our findings cannot be entirely attributed to the intervention, as we did not assess training motivation, and climbing or bouldering enjoyment prior to the intervention.

Nonetheless, we can state that lower-grade to advanced female climbers who had not engaged in any kind of climbing-specific training before, expressed relatively high exercise enjoyment during the intervention and relatively high intrinsic training motivation after the study. This is an important finding for injury prevention and performance improvement in recreational climbing and may serve as a guide for climbing and bouldering centers.

In order to allow for sufficient study power (80%, BF $\geq$ 10) Brysbaert [37] recommends at least 52 participants for 2 x 2 hypothesis testing. Due to feasibility constraints and limited participant availability, we were only able to include 27 participants. This indicates that the study is underpowered and future studies should aim to include larger samples.

## Conclusion

This study investigated the effects of on- and off-the wall climbing training on climbing performance, climbing specific strength, and training experience in lower-level to advanced female climbers.

While we were not able to find any differences between the groups in bouldering performance, and climbing-specific strength, individual-response analysis revealed that on-the-wall training tended to enhance climbing technique, while off-the-wall showed a trend toward positive effects on climbing-specific strength. Our findings further suggest that lower-grade to advanced female climbers, previously unengaged in climbing-specific training, exhibit high exercise enjoyment and intrinsic training motivation after training supervision for both on- and off-the-wall training.

When deciding to train, athletes of this population should thus rely on personal preferences and don't need to engage in uncomfortable or painful training.

Future studies, should include larger samples and higher-level climbers, and focus on longer intervention periods. Furthermore, follow up questionnaires should be incorporated to check for possible intention-action gaps.Walls with built-in lights and adjustable angles have been developed to display a variety of boulder problems. With the standardisation of hand and foot holds and consensus-based difficulty gradings, these walls represent a potentially useful tool for standardising performance measures, quantifying and tailoring training programmes, and adapting difficulty and style to all levels of climbers.

## Supporting information

**S1 Fig. Boulder A-E.**
(TIF)

**S2 Fig. Individual training responses to performance measures.**
(TIF)

**S3 Fig. Individual training responses to strength measurements.**
(TIF)

**S4 Fig. Posterior plots of the consistency analysis of the different scales and subscales used.**
Original values and values in case one item was dropped. PACES–physical activity enjoyment scale, IMI–Intrinsic Motivation Inventory.
(TIF)

**S5 Fig. Number of hours spent on other sports during the time of the intervention by individuals of each group.** WT–off-the-wall training group, ST–on-the-wall training group, CG–control group.
(TIF)

**S1 Table. Tests and corresponding measures.** RPE–rate of perceived exertion, RPD–rate of perceived discomfort, PACES–physical activity enjoyment scale, EES–exercise enjoyment scale, FS–feeling scale, IMI–Intrinsic Motivation Inventory.
(PDF)

**S2 Table. Performance assessment tool.**
(PDF)

**S3 Table. Intrinsic motivation inventory.** Interest/enjoyment–items 1, 2, 9, 11, 13, 20, 23; effort/importance–items 5, 6, 12, 16, 21; pressure/tension–items 4, 7, 14, 17, 19; value usefulness–items 3, 8, 10, 15, 18, 22, 24.
(PDF)

**S4 Table. Number of completed training sessions and hours spent climbing and bouldering during the five-week intervention by each participant.** WT–off-the-wall training group, ST–on-the-wall training group, CG–control group.
(PDF)

**S5 Table. Training diary (CG did not fill in the last column).**
(PDF)

**S6 Table. Pre- and post-test results for all strength and performance tests.** CG–control group, WT–on -the-wall training group, ST–off-the -wall training group.
(PDF)

**S7 Table. Inter-rater reliability.** $BF_{10}$ –Bayes factor (evidence for the alternative hypothesis relative to the null hypothesis/null model).
(PDF)

**S8 Table. Intra-rater reliability.** $BF_{10}$ –Bayes factor (evidence for the alternative hypothesis relative to the null hypothesis/null model).
(PDF)

**S9 Table. Individual development from pre- to post-test across groups and tests.** A decrease in the total number of attempts was rated as a performance increase.––performance decrease, ○ –no performance increase, +–performance increase, WT–off-the-wall training group, ST–on-the-wall training group, CG–control group.
(PDF)

**S10 Table. Correlation between items of different scales.** RPE–Rate of perceived exertion, RPD–rate of perceived discomfort, FS–feeling scale, EES–exercise enjoyment scale, PACES–physical activity enjoyment scale, IMI–Intrinsic Motivation Inventory, CG–control group, WT–on-the-wall training group, ST–off-the-wall training group, SD–standard deviation.
(PDF)

**S11 Table. Correlation between items of different scales.** RPD–rate of perceived discomfort, EES–exercise enjoyment scale, FS–feeling scale, PACES–physical activity enjoyment scale,

BF10 –Bayes factor (evidence for the alternative hypothesis relative to the null hypothesis.
(PDF)

**S12 Table. Raw dataset.**
(PDF)

## Acknowledgments

The authors express their gratitude to Prof. Dr. Atle Hole Saeterbakken, Prof. Dr. Josef Wie-meyer, and Dr. Christian Simon, whose critical insights and constructive feedback have significantly enriched this study. Additionally, the authors thank all climbers who participated in the study.

## Author Contributions

**Conceptualization:** Kaja Langer, Nicolay Stien.

**Formal analysis:** Kaja Langer.

**Investigation:** Kaja Langer.

**Methodology:** Kaja Langer.

**Resources:** Vidar Andersen.

**Writing – original draft:** Kaja Langer.

**Writing – review & editing:** Vidar Andersen, Nicolay Stien.

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
