## [Decision Letter · Decision Letter 0]

13 May 2024

PONE-D-24-12063The Effects of Five Weeks of Climbing Training, On and Off the Wall, on Climbing Specific Strength, Performance, and Training Experience in Female ClimbersPLOS ONE

Dear Dr. Langer,

Thank you for submitting your manuscript to PLOS ONE. After careful consideration, we feel that it has merit but does not fully meet PLOS ONE’s publication criteria as it currently stands. Therefore, we invite you to submit a revised version of the manuscript that addresses the points raised during the review process.

We look forward to receiving your revised manuscript.

Kind regards,

Mehrnaz Kajbafvala, Ph.D

Academic Editor

PLOS ONE

Journal Requirements:

"There were no professional relationships with companies or manufacturers that will benefit from the results of this study. There was no specific financial support for the preparation of this study. The authors have no potential conflicts of interest that are directly related to the study and the journal."

Reviewers' comments:

Reviewer's Responses to Questions

**Comments to the Author**

1. Is the manuscript technically sound, and do the data support the conclusions?

Reviewer #1: Yes

Reviewer #2: Yes

2. Has the statistical analysis been performed appropriately and rigorously? 

Reviewer #1: Yes

Reviewer #2: I Don't Know

3. Have the authors made all data underlying the findings in their manuscript fully available?

Reviewer #1: Yes

Reviewer #2: Yes

4. Is the manuscript presented in an intelligible fashion and written in standard English?

Reviewer #1: Yes

Reviewer #2: Yes

5. Review Comments to the Author

Reviewer #1: 1-Please mark significances in diagrams.

2-Please omit lines of tables and mark alternative rows by colors...................................................................................................

Reviewer #2: Title: The type of study should be written in the title

Main text:

The introduction should be shortened. The article as a whole is very long and this amount of expansion and dispersion hinders the transmission of the main goal and the valuable findings of the research.

How did you calculate the sample size (31 cases)?

Randomization is not adequately explained.

RCTs should be registered and the registration number should be inserted.

Abbreviations should be written below each table.

There are too many tables. Most of the tables should be moved to the supplementary file (especially tables 2-7).

Do the graphs replicate the data in the tables? If the answer is yes, what was the need to repeat? It is better not to repeat data in results.

In general, the article is very long, scattered, and of course valuable data, and specialized in sports, which seems to be more suitable for specialized sports journals.

6. PLOS authors have the option to publish the peer review history of their article (what does this mean?). If published, this will include your full peer review and any attached files.

Reviewer #1: **Yes: **soheil mansour sohani

Reviewer #2: No

---

## [Author Response · Author response to Decision Letter 0]

27 May 2024

Reviewer #1: 

Thank you for your comments! Especially the first has helped to increase the quality of our paper.

• Please mark significances in diagrams. – Significance was included in figures 7 and 8. As MANCOVAs were calculated for strength and performance tests, no differences were determined for the groups regarding the individual tests. Significance can therefore not be added to figures 5 and 6.

• Please omit lines of tables and mark alternative rows by colors – This was implemented as proposed.

Reviewer #2: 

Thank you for your thorough review! In particular, the readability of our paper was greatly improved through your feedback.

• The type of study should be written in the title – The title was changed accordingly.

• The introduction should be shortened. The article as a whole is very long and this amount of expansion and dispersion hinders the transmission of the main goal and the valuable findings of the research. – The introduction was shortened.

• How did you calculate the sample size (31 cases)? – We added sufficient sample size as proposed by Brysbaert to the study design section. The town where the study was conducted is very small and has a very small climbing community. Therefore, we were not able to meet the criteria, but tried to recruit as many subjects as possible. Study power is discussed in the discussion.

• Randomization is not adequately explained. – The randomization process is described in detail in the methods section. We have highlighted the respective section with a comment in the tracked manuscript.

• RCTs should be registered and the registration number should be inserted. – You are right, the study should have been registered in advance to ensure additional quality. This was neglected by the authors. Nevertheless, the study was thoroughly evaluated by an ethics commission. Unfortunately, registering a study after it has been completed is not in line with ICMJE recommendations, which is why we have decided against registering the study retrospectively. However, we will take the comment into account and consider preregistration in future studies.

• Abbreviations should be written below each table. – This was implemented as proposed.

• There are too many tables. Most of the tables should be moved to the supplementary file (especially tables 2-7). – Some of the tables suggested and a few others (5 in total) were moved to the supplementary file. The total number of tables was reduced to 11

• Do the graphs replicate the data in the tables? If the answer is yes, what was the need to repeat? It is better not to repeat data in results. – Tables repeating the data were moved to the supplementary material. It was important for us to provide the exact data but we see the point in not repeating the results in the paper itself.

• In general, the article is very long, scattered, and of course valuable data, and specialized in sports, which seems to be more suitable for specialized sports journals. – Climbing research is still very young. PLOS ONE, in particular, has published many important findings in the field of climbing and thus supports a basis for research in this area. We have decided to submit our paper here in order to reach the climbing community in research and practice in the best possible way, and would like to stick to this submission for the reasons mentioned above.

---

## [Decision Letter · Decision Letter 1]

17 Jun 2024

The Effects of Five Weeks of Climbing Training, On and Off the Wall, on Climbing Specific Strength, Performance, and Training Experience in Female Climbers – A Randomized Controlled Trial

PONE-D-24-12063R1

Dear Dr. Kaja Langer

We’re pleased to inform you that your manuscript has been judged scientifically suitable for publication and will be formally accepted for publication once it meets all outstanding technical requirements.

Kind regards,

Mehrnaz Kajbafvala, Ph.D

Academic Editor

PLOS ONE

Additional Editor Comments (optional):

Reviewers' comments:

Reviewer's Responses to Questions

**Comments to the Author**

1. If the authors have adequately addressed your comments raised in a previous round of review and you feel that this manuscript is now acceptable for publication, you may indicate that here to bypass the “Comments to the Author” section, enter your conflict of interest statement in the “Confidential to Editor” section, and submit your "Accept" recommendation.

Reviewer #1: All comments have been addressed

Reviewer #2: All comments have been addressed

2. Is the manuscript technically sound, and do the data support the conclusions?

Reviewer #1: Yes

Reviewer #2: Partly

3. Has the statistical analysis been performed appropriately and rigorously? 

Reviewer #1: Yes

Reviewer #2: I Don't Know

4. Have the authors made all data underlying the findings in their manuscript fully available?

Reviewer #1: Yes

Reviewer #2: Yes

5. Is the manuscript presented in an intelligible fashion and written in standard English?

Reviewer #1: Yes

Reviewer #2: Yes

6. Review Comments to the Author

Reviewer #1: Congratulations.

Your manuscript is publishable...................................................................................................................................................................................

Reviewer #2: Thank you for your response.I was not convinced about the sample size. Anyway, I have no other comment. Good luck

7. PLOS authors have the option to publish the peer review history of their article (what does this mean?). If published, this will include your full peer review and any attached files.

Reviewer #1: **Yes: **Soheil Mansour Sohani

Reviewer #2: No

---

## [Editor Report · Acceptance letter]

27 Jun 2024

PONE-D-24-12063R1 

PLOS ONE

Dear Dr. Langer, 

I'm pleased to inform you that your manuscript has been deemed suitable for publication in PLOS ONE. Congratulations! Your manuscript is now being handed over to our production team.

Kind regards, 

on behalf of

Dr. Mehrnaz Kajbafvala 

Academic Editor

PLOS ONE